# UniCM: A Unified Consistency Model For Efficient Multimodal Generation and Understanding

## Abstract

Consistency models (CMs) have shown promise in the efficient generation of both image and text. This raises the natural question of whether we can learn a unified CM for efficient multimodal generation (e.g., text-to-image) and understanding (e.g., image-to-text). Intuitively, such a model could be acquired by applying the consistency distillation (CD) to existing unified multimodal models. However, the key challenge is establishing a unified denoising perspective for both image and text generation, which is essential for establishing the consistency mapping. To tackle this, at the representation level, we advocate for discrete tokens for both modalities to best preserve language modeling capabilities. Critically, instead of defining the text denoising trajectory via recent discrete diffusion language modeling principles, we specify it using the parallel decoding trace of an autoregressive language model, benefiting from the latter's superior performance in general text generation tasks. The denoising trajectory of image tokens adheres to standard discrete diffusion. We train our unified consistency models (UniCMs) on these combined multimodal trajectories simultaneously with a unified objective. We introduce a trajectory segmentation strategy to improve the training convergence. Empirically, in text-to-image generation, UniCMs outperform SD3 on GenEval and Image Reward, while requiring only approximately $1/8$ of the sampling time. Meanwhile, in image-to-text generation, UniCMs surpass Show-o on the MMMU benchmark while being $1.5\times$ faster at long-sequence generating speed.

## 1 Introduction

Consistency models (CMs) (Song et al., 2023) have made significant achievements in efficient content generation across modalities. For image generation, CMs have revolutionized diffusion models, synthesizing high-fidelity images with few sampling steps (Song et al., 2023; Luo et al., 2023; Song & Dhariwal, 2023; Ren et al., 2024; Xie et al., 2024b; Wang et al., 2024a). Recently, CMs have been extended to text generation, realizing inference acceleration up to 3 times (Kou et al., 2024a). Naturally, this raises an important question: *can such advances in different modalities lead to a unified consistency model capable of efficiently understanding and generating cross-modal data?*

Given the recent progress on unified multimodal generation and understanding models (Team, 2024b; Zhou et al., 2024; Wang et al., 2024b; Xie et al., 2024a), it is intuitive to apply consistency distillation (CD) (Song et al., 2023) to them to acquire unified consistency models. However, this cannot be implemented trivially due to a dilemma—the consistency mapping needs to be defined on a denoising-style generation trajectory, but how to establish a unified denoising perspective that encompasses both text and image generation remains an open challenge.

To address this, this paper introduces UniCM, a unified consistency model. We advocate discrete tokenization for both modalities at the representation level, which preserves language modeling ability. Thus, the core problem boils down to constructing a unified discrete denoising trajectory for the generation of both image and text tokens. For the former, we follow the typical masked diffusion paradigm (e.g., Muse (Chang et al., 2023), MaskGit (Chang et al., 2022), MagVit (Yu et al., 2023), and Show-o (Xie et al., 2024a)). For the latter, we suggest specifying the denoising trajectory with the parallel decoding trace of an autoregressive (AR) language generation process,

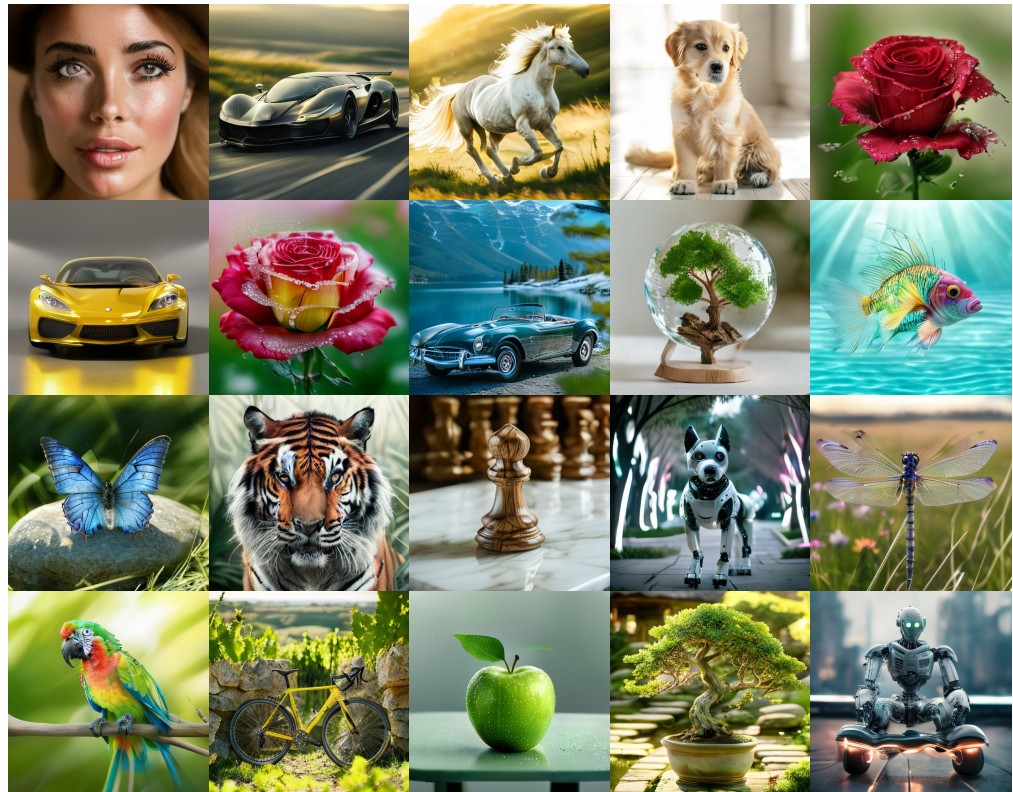

Figure 1: **512 × 512 images generated by UniCMs.** All images are generated by UniCMs in 4 sampling steps without reliance on classifier-free guidance (Ho & Salimans, 2021).

given the success of consistency LLMs (CLLMs) (Kou et al., 2024a). We bypass the recent discrete diffusion language models (Nie et al., 2025; Ye et al., 2025; Nie et al., 2024; Gong et al., 2024) due to weaker performance and limited multimodal applicability compared to AR ones.

With such multimodal trajectories, we train the unified consistency models (UniCMs) using a unified objective. Specifically, UniCMs are pushed to consistently map any point on the trajectory to the same endpoint to enable fast-forward generation. We introduce a trajectory segmentation strategy (Heek et al., 2024; Zheng et al., 2024; Xie et al., 2024b) in which distillation is applied to each segment of the complete generation trajectory to improve convergence. We also design regularizations to ensure the training stability. Conceptually, our approach constitutes an empirical generalization of the original CMs (Song et al., 2023) to discrete denoising trajectories and establishes a cross-modal extension of CLLMs.

Given that Show-o (Xie et al., 2024a) can perform AR generation for text tokens and mask diffusion generation for image tokens, we opt to leverage it to collect text-to-image denoising trajectories on COCO 2017 (Lin et al., 2014) and image-to-text ones on LLaVA instruction tuning dataset (Liu et al., 2024d). We then initialize UniCMs with Show-o and perform fine-tuning on such trajectories. This training lasts for 36 hours on 8 A100-40GB GPUs. For text-to-image generation, UniCMs outperform SD3 (Esser et al., 2024) on GenEval (Ghosh et al., 2023), Image Reward (IR) (Li et al., 2024b), and CLIP Score (CS) (Hessel et al., 2022), while requiring only approximately $1/8$ of time. For image-to-text generation, UniCMs surpass Show-o on the MMMU (Yue et al., 2024) benchmark while being approximately $1.5\times$ faster on the captioning tasks like NoCaps (Agrawal et al., 2019).

In summary, our main contributions are as follows:

- We propose UniCMs, a novel unified consistency model family, which enables efficient multimodal understanding and generation within a single backbone architecture.
- For text-to-image generation, UniCMs outperform SD3 (Esser et al., 2024) while requiring only $1/8$ of time. For image-to-text generation, UniCMs surpass Show-o on MMMU (Yue et al., 2024) while being approximately $1.5\times$ faster on NoCaps (Agrawal et al., 2019).

## 2 RELATED WORK

**Unified Models.** Early generative models typically specialized in either text-conditioned image generation (Rombach et al., 2022; Podell et al.; Song et al., 2020; Chen et al., 2023; 2024; Li et al., 2024c; Yang et al., 2024; Sun et al., 2024) or vision–language understanding (Liu et al., 2024c; Lin et al., 2024; Liu et al., 2024d;b; Li et al., 2024a; Zhu et al., 2024; Bai et al., 2023; Ye et al., 2024; Zhu et al., 2023), handling only one direction of multimodal interaction. To address this, unified multi-modal models (Wu et al., 2023a; Zhao et al., 2024; Chern et al., 2024; Dong et al., 2023; Wu et al., 2024) have been proposed to support both image and text tasks. For example, Chameleon (Team et al., 2023) and Emu3 (Wang et al., 2024b) autoregressively generate text and image tokens, while Transfusion (Zhou et al., 2024) combines autoregressive and continuous diffusion methods. Similarly, Show-o (Xie et al., 2024a) uses autoregressive text generation with discrete diffusion for images. Although these unified models mark progress toward versatile multimodal systems, their reliance on iterative generation still incurs high computational cost.

**Consistency Models (CMs).** CMs have gained attention for generating high-quality outputs efficiently. First proposed for continuous diffusion models (Song et al., 2023; Luo et al., 2023), they introduce trajectory consistency: mapping any two points along a sampling trajectory to a shared endpoint (Song et al., 2023). This property lets the model skip intermediate steps and directly predict the endpoint, enabling high-quality generation in far fewer steps—sometimes even one. Building on this idea, multi-step CMs segment trajectories and enforce consistency within each segment (Zheng et al., 2024; Heek et al., 2024; Xie et al., 2024b; Wang et al., 2024a). While research has focused on continuous domains, the principle has also been applied to discrete diffusion models (Hayakawa et al., 2024), though with limited efficiency gains. Consistency distillation has further been adapted to accelerate large language models (LLMs) by applying similar objectives to iterative text generation (Kou et al., 2024a). However, these efforts have largely concentrated on continuous diffusion models, primarily for image generation, or on purely text-based models addressing single tasks. To date, unified consistency models remain largely unexplored.

## 3 METHOD

This section presents unified consistency models (UniCMs) for efficient multimodal generation and understanding. We first review existing approaches on unified models and then provide insights on how to establish a unified denoising trajectory for learning UniCMs. We also elaborate on the unified CD loss as well as a suite of strategies to improve the model training.

### 3.1 PRELIMINARY: UNIFIED MULTIMODAL MODELS

Unified multimodal modeling aims to process both textual and visual modalities within a compact model for joint generation (Team et al., 2023; Wang et al., 2024b; Team, 2024b). Typically, the architecture includes a transformer backbone, an encoder and decoder for images, and a text tokenizer. The image encoder converts an input image into patch-wise tokens $\mathbf{u} = \{u_1, \ldots, u_m\}$, where $m$ is the number of patches and $u_i$ can be continuous vectors or discrete indices derived from vector quantization (Van Den Oord et al., 2017). The text tokenizer encodes text into $n$ discrete tokens $\mathbf{v} = \{v_1, \ldots, v_n\}$. The unified model then characterizes the text-to-image (T2I) and image-to-text (i.e., multimodal understanding, MMU) relationships simultaneously with the shared transformer backbone. In particular, the backbone predicts image and text tokens, which are then decoded by the image decoder and detokenized, respectively, to obtain images and text.

Unified models typically generate text tokens $\mathbf{v}$ autoregressively, a consequence of language's discrete and sequential nature. Formally, the learning objective is Next Token Prediction (NTP):

$$\mathcal{L}_{\text{NTP}} := \sum_i \log p_\theta(v_i | v_1, \cdots, v_{i-1}, \mathbf{u}), \tag{1}$$

where $\theta$ denotes learnable parameters and $p_\theta$ refers to model likelihood.

Based on how $\mathbf{u}$ are produced, existing approaches can be categorized into three main classes:

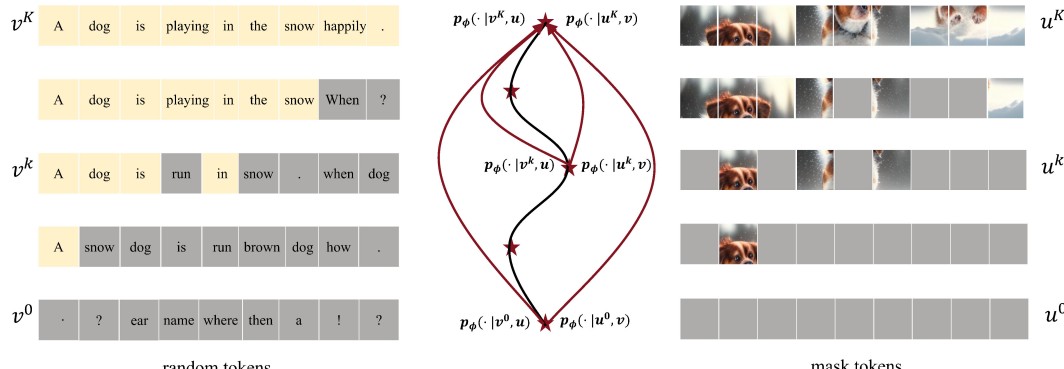

Figure 2: **Illustration of the unified denoising perspective of text and image generation.** As shown, the trajectories both display a denoising pattern. The black line denotes the unified abstraction of the multimodal trajectory, and the red lines illustrate the objective of UniCMs—to map an arbitrary point on the sampling trajectory to the same endpoint for both text and image generation. Note that we omit the trajectory segmentation strategy in the training process for brevity.

- Autoregressive generation (Sun et al., 2024; Ma et al., 2024) of $\mathbf{u}$, where $\mathbf{u}$ are discrete, as seen in models like Emu3 (Wang et al., 2024b), Chameleon (Team, 2024b), LWM (Liu et al., 2024a), etc.
- Discrete diffusion (Chang et al., 2022; 2023; Yu et al., 2023) generation of $\mathbf{u}$, which also relies on discrete $\mathbf{u}$ and is known as mask diffusion, exemplified by Show-o (Xie et al., 2024a).
- Gaussian diffusion (Ho et al., 2020; Song et al., 2020; Luo et al., 2023) generation of $\mathbf{u}$, where $\mathbf{u}$ are continuous vectors, as demonstrated in Transfusion (Zhou et al., 2024).

Despite the promise of multimodal generation, unified models remain slow, especially in T2I. For instance, Emu3 takes over a minute to produce a $512 \times 512$ image on an NVIDIA 4090 GPU due to long image tokens (e.g., 4096). Diffusion models like Show-o and Transfusion improve efficiency but still trail specialized T2I models (Ren et al., 2024; Sauer et al., 2024). Similarly, image-to-text also demand acceleration, as outputs can be lengthy, e.g., image captioning (Plummer et al., 2017; Young et al., 2014) and multimodal chain-of-thought reasoning (Shen et al., 2025; Feng et al., 2025).

**Efficient Unified Generation and Understanding by CMs.** CMs enable efficient generation in both image (Song et al., 2023; Luo et al., 2023) and text (Kou et al., 2024a), providing a unified framework. Given a denoising trajectory, CMs map any two points to a common endpoint for fast-forward generation. Thus, unified CMs require a shared trajectory across modalities. When aligning discrete image tokens with text tokens, the key challenge is defining a unified discrete denoising trajectory.

### 3.2 A UNIFIED DENOISING PERSPECTIVE FOR THE GENERATION OF IMAGE AND TEXT

**Denoising Trajectory for Image.** A natural approach to obtain a discrete denoising trajectory for image tokens $\mathbf{u}$ is through discrete diffusion modeling. Typically, the process begins with a sequence of $m$ fully masked image tokens $\mathbf{u}^0 := \{u_1^0, \dots, u_m^0\}$, with the mask ratio progressively decreasing to 0 over $K$ iterative steps. Specifically, in the $k$-th step, given the sequence $\mathbf{u}^k$, let $M_k$ be the set of indices of masked tokens within $\mathbf{u}^k$. The model first predicts the tokens for all masked positions $i \in M_k$ to get an intermediate sequence $\bar{\mathbf{u}}^{k+1}$ as follows:

$$\bar{u}_i^{k+1} = \begin{cases} \arg\max_u p_\theta(u_i = u | \mathbf{u}^k, \mathbf{v}), & \text{if } i \in M_k \\ u_i^k, & \text{if } i \notin M_k \end{cases} \quad (2)$$

where $\mathbf{v}$ denotes the text condition and $p_\theta$ is abused to denote a T2I model that employs masked diffusion modeling on images (e.g., Show-o (Xie et al., 2024a), Muse (Chang et al., 2023), and Meissonic (Bai et al., 2024)). Then, the model re-masks low-confidence generations in $\bar{\mathbf{u}}^{k+1}$ according to the mask ratio schedule, yielding $\mathbf{u}^{k+1}$. The resultant trajectory $\{\mathbf{u}^0, \mathbf{u}^1, \dots, \mathbf{u}^K\}$ is visualized in Figure 2.

**Denoising Trajectory for Text.** To obtain text denoising trajectories, we consider two approaches: (1) leveraging recent discrete diffusion-based language generation methods (Ye et al., 2025; Nie et al., 2025) or (2) utilizing the parallel decoding trajectories derived from an AR language generation process, as suggested by CLLMs (Kou et al., 2024a). Given the slightly inferior performance and limited application in processing multimodal inputs of diffusion language models compared to AR ones, we opt for the latter.

Technically, starting from a sequence of $n$ randomly initialized text tokens, denoted as $\mathbf{v}^0 := \{v_1^0, \ldots, v_n^0\}$, the parallel decoding process iteratively refines the token sequence until a fixed point. At $k$-th iteration, the refinement corresponds to simultaneously solving the following $n$ problems:

$$
\begin{aligned}
v_1^{k+1} &= \arg\max_v p_\theta(v|\mathbf{u}), \\
v_2^{k+1} &= \arg\max_v p_\theta(v|v_1^k, \mathbf{u}), \\
&\ldots \\
v_n^{k+1} &= \arg\max_v p_\theta(v|v_1^k, \ldots, v_{n-1}^k, \mathbf{u}),
\end{aligned}
\tag{3}
$$

where $p_\theta$ is abused for an image-to-text AR model. In fact, these problems can be solved simultaneously with only one forward pass using a causal attention mask, which takes roughly identical time as decoding one new token. Note that the greedy sampling strategy is used here. Abusing $K$ to denote the number of iterations to reach the fixed point $\mathbf{v}^K$, it is easy to see $K \le n+1$ because there is at least one token being correctly predicted in each iteration.[1] Refer to Figure 2 for a visualization of the sampling trajectory $\{\mathbf{v}^0, \ldots, \mathbf{v}^K\}$, which displays a gradual denoising pattern.

## 3.3 TRAINING OF UNICMS

Based on the foregoing, text trajectories can be sourced from AR image-to-text models (like LLaVA (Liu et al., 2023), Qwen-VL-chat (Bai et al., 2023), Show-o (Xie et al., 2024a)), and image trajectories from mask diffusion T2I models (like Show-o (Xie et al., 2024a), Muse (Chang et al., 2023), and Meissonic (Bai et al., 2024)). Given Show-o's ability to fulfill both roles, we favor it in our current work. Furthermore, this preference naturally extends to initializing UniCMs with Show-o's architecture and parameters when training on its trajectories, facilitating a smoother cold start. Letting $p_\phi$ denote the UniCMs to learn, we elaborate on the algorithmic details below.

**Unified Training Objective.** The consistency loss on image trajectories is:

$$
\mathcal{L}_c^u = \mathbb{E}_{k \sim \mathcal{U}(0,K)} d\left(p_{\phi^-}(\cdot|\mathbf{u}^K, \mathbf{v}), p_\phi(\cdot|\mathbf{u}^k, \mathbf{v})\right),
\tag{4}
$$

where $\phi^-$ denotes stopping gradient backpropagation for stable training (Song et al., 2023) and $d$ indicates a divergence measure. For $\mathcal{L}_c^u$, $d$ aggregates the KL divergence between categorical prediction distributions over the masked image tokens. The consistency loss on text trajectories can be similarly defined:

$$
\mathcal{L}_c^v = \mathbb{E}_{k \sim \mathcal{U}(0,K)} d\left(p_{\phi^-}(\cdot|\mathbf{u}, \mathbf{v}^K), p_\phi(\cdot|\mathbf{u}, \mathbf{v}^k)\right),
\tag{5}
$$

where $d$ aggregates over the positions where the two prediction distributions differ. These losses, $\mathcal{L}_c^u$ and $\mathcal{L}_c^v$, are global consistency losses for image and text trajectory (mapping to their respective endpoints $\mathbf{u}^K$ and $\mathbf{v}^K$), empirically superior to local losses for discrete denoising trajectories (Kou et al., 2024a). Conceptually, our objective forms an empirical generalization of the original CMs defined on the ODE trajectories and a cross-modal extension of CLLMs (Kou et al., 2024a).

**Trajectory Segmentation.** We empirically ascertain that imposing long-range consistency may introduce unnecessary learning challenges, potentially impeding model convergence and ultimately limiting the model's inference efficiency. Inspired by previous work (Heek et al., 2024; Zheng et al., 2024; Xie et al., 2024b), we design a segmentation strategy for the collected discrete multimodal sampled trajectories, enforcing consistency and regularization constraints in specific regions between points within a segment and segment endpoints. More details about the trajectory segmentation can be found in Appendix E.

As the training proceeds, the trajectories of UniCMs may deviate significantly from the original collected multimodal trajectories. Thus, persisting in utilizing the original trajectory for distillation

---

[1] By correctness, we mean the generated tokens equal to those generated by regular AR decoding.

| Type | Model | Res. | Steps | GenEval ↑ | HPS ↑ | IR ↑ | CS ↑ | Time (s) ↓ |
|------|-------|------|-------|-----------|-------|------|------|------------|
| Gen. Only | Emu3-Gen (Wang et al., 2024b) | 512 | 4096 | 0.540 | - | - | - | 309.51 |
| | SDXL (Podell et al., 2023) | 1024 | 50 | 0.550 | 0.267 | 0.698 | 0.312 | 6.88 |
| | SDXL-Turbo (Sauer et al., 2024) | 512 | 1 | 0.551 | 0.273 | 0.759 | 0.315 | 0.27 |
| | SD3 (Esser et al., 2024) | 512 | 24 | 0.620 | 0.275 | 0.787 | 0.308 | 1.33 |
| | Hyper-SD3 (Ren et al., 2024) | 1024 | 4 | 0.458 | 0.266 | 0.649 | 0.308 | 1.19 |
| Und. & Gen. | Show-o (Xie et al., 2024a) | 512 | 16 | **0.674** | **0.277** | **0.992** | **0.318** | 1.39 |
| | | 512 | 8 | 0.578 | 0.257 | 0.672 | 0.313 | 0.76 |
| | Transfusion (Zhou et al., 2024) | 256 | 250 | 0.630 | - | - | - | - |
| | Chameleon (Team, 2024a) | 512 | 1024 | 0.430 | - | - | - | 19.24 |
| | Orthus (Team, 2024a) | 512 | 1024 | 0.580 | - | - | - | 239.90 |
| | UniCMs | 512 | 8 | 0.638 | 0.273 | 0.963 | **0.318** | 0.33 |
| | | 512 | 4 | 0.625 | 0.269 | 0.934 | **0.318** | 0.17 |
| | | 512 | 2 | 0.557 | 0.247 | 0.680 | 0.312 | **0.09** |

Table 1: **Comparison of model performance for T2I task.** For the "Und. & Gen." panel, best results are shown in **bold** and second best results are underlined.

purposes could constrain the ultimate acceleration effect. We propose to regenerate multimodal denoising trajectories using the consistency model obtained in past stages. In this training stage, we also halve the number of segments of the trajectory to achieve better acceleration. Doing so encourages the final UniCMs to learn consistency mapping over long distances.

**Regularization.** Training UniCMs with only consistency loss in discrete multimodal denoising can lead to trivial convergence (e.g., identical outputs for varied inputs). To prevent this, we add regularizations for both modalities. For text, $p_\phi$ must fit endpoint tokens $\mathbf{v}^K$ via an NTP objective. For images, we observe that the prediction logits of recovered image tokens contain rich information (e.g., easy-to-difficult hierarchies), so record them at each sampling step during trajectory collection (detailed in Appendix D, Figure 7). Then, we use the logits as targets to regularize $p_\phi(\cdot|\mathbf{u}^k, \mathbf{v})$.

We use $\mathcal{L}_{REG}^v$ and $\mathcal{L}_{REG}^u$ to represent these two regularizations respectively. The total loss is

$$\mathcal{L} = \mathcal{L}_c^u + \alpha\mathcal{L}_c^v + \beta\mathcal{L}_{REG}^u + \gamma\mathcal{L}_{REG}^v, \tag{6}$$

where $\alpha$, $\beta$ and $\gamma$ are the trade-off coefficients to balance the different losses.

**Sampling Strategy.** We find that for the learned UniCMs with few sampling steps, there is significantly higher uncertainty in the prediction distribution of the mask tokens. We empirically identify that incorporating the top-k sampling strategy, which is widely used in language models, can alleviate this issue, substantially improving the sampling quality in 2-4 steps (see Table 3).

## 4 EXPERIMENTS

This section evaluates on T2I generation and MMU tasks to inspect the efficacy of UniCMs.

### 4.1 IMPLEMENTATION DETAILS

**Datasets.** The captions from the training split of COCO 2017 (Lin et al., 2014) are used to generate text-to-image denoising trajectories. The LLaVA instruction tuning dataset (Liu et al., 2024d) is employed to collect image-to-text denoising trajectories. Besides, the RefinedWeb text dataset (Penedo et al., 2023) is incorporated to preserve the model's language modeling capabilities through autoregressive objective.

**Training Details.** We train UniCMs at two resolutions: results at 512 are in the main text, while 256 results and details appear in Appendix F. Training has two stages. For 512 resolution, stage one collects image trajectories with classifier-free guidance (CFG) (Ho & Salimans, 2021) scale 15 and $K = 32$, splitting each trajectory into 8 segments to train UniCMs*. Stage two collects trajectories from UniCMs* with CFG scale 1.75, $K = 16$, and 4 segments. Text trajectories are collected analogously. We use parallel decoding, generating 16 tokens per block to form long text, which

| Type | Method | Param | TPS ↑ | POPE ↑ | SQA ↑ | MMMU ↑ | NoCaps ↑ | Flickr30k ↑ |
|---|---|---|---|---|---|---|---|---|
| Und. Only | Emu3-Chat (Wang et al., 2024b) | 8B | 13.8 | 85.2 | - | 31.6 | - | - |
| | Qwen-VL-chat (Bai et al., 2023) | 7B | 26.8 | - | - | 35.9 | 15.4 | 9.2 |
| | InstructBLIP (Dai et al., 2023) | 7B | - | 78.9 | 31.2 | 28.1 | 30.4 | 24.8 |
| Und. & Gen. | Show-o (Xie et al., 2024a) | 1.3B | 40.3 | **83.2** | 34.9 | 24.6 | **29.4** | **24.9** |
| | Orthus (Kou et al., 2024b) | 7B | 7.6 | 79.6 | - | **28.2** | - | - |
| | Chameleon (Team, 2024a) | 7B | 11.47 | 77.8 | - | 26.7 | - | - |
| | UniCMs | 1.3B | **61.1** | 78.4 | **37.1** | 26.3 | 26.5 | 22.2 |

Table 2: **Comparison of MMU performance on multiple benchmarks.** Note that SQA refers to ScienceQA-IMG. POPE and MMMU measure question-answering ability, while Flickr30K and NoCaps evaluate the ability of image description.

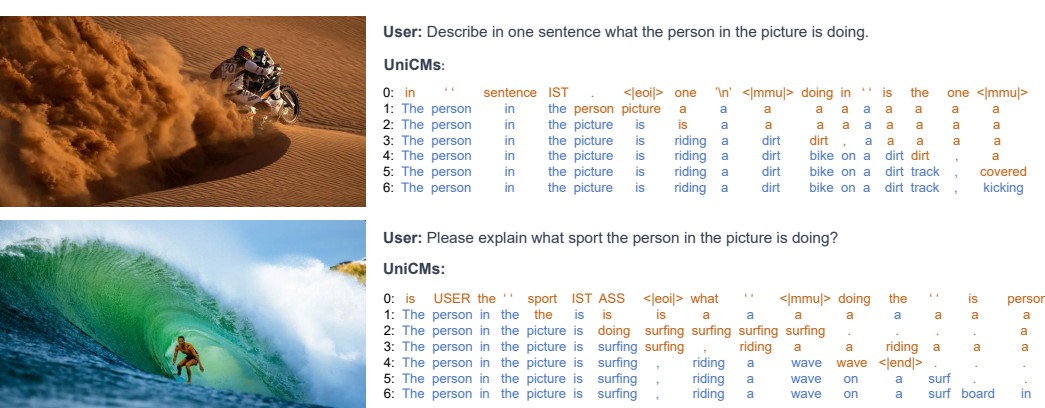

Figure 3: **The text sampling trajectory of UniCMs in MMU cases.** UniCMs realize acceleration by predicting multiple successive tokens in one iteration and correctly guessing the later tokens.

accelerates generation while preserving modeling ability (Kou et al., 2024a). Multimodal trajectories are collected deterministically in both stages for stability, though UniCMs remain compatible with stochastic sampling (Table 3). Loss coefficients are set as $\alpha = 10$ according to the relative values of the losses, $\beta = 40$, and $\gamma = 200$ (according to Table 5). Each stage is trained with AdamW on 8 A100 GPUs for 18 hours at constant learning rate $10^{-5}$. At inference, UniCMs run without CFG, further reducing computation.

## 4.2 MAIN RESULTS

**Benchmarks.** We evaluate UniCMs in the T2I task on Human Preference Dataset v2 (HPD) (Wu et al., 2023b), using metrics including Human Preference Score v2 (HPS) (Wu et al., 2023b), ImageReward (IR) (Xu et al., 2023), and CLIP Score (CS) (Hessel et al., 2022). In addition, we conduct a comprehensive evaluation of UniCMs on the GenEval (Ghosh et al., 2023) benchmark. For MMU, we assess UniCMs on the image description benchmarks Flickr30K (Plummer et al., 2017; Young et al., 2014) and NoCaps (Agrawal et al., 2019) measured by the *METEOR* (Banerjee & Lavie, 2005) metric and calculate the accuracy on question answering benchmarks, including POPE (Li et al., 2023), ScienceQA (Lu et al., 2022), and MMMU (Yue et al., 2024).

**Baselines.** For T2I, we compare UniCMs with typical unified models (e.g., Transfusion (Zhou et al., 2024), Orthus (Team, 2024a), Show-o (Xie et al., 2024a)) and some outstanding image generation models (e.g., Emu3-Gen (Wang et al., 2024b), SD-XL (Podell et al.) and SD3 (Esser et al., 2024)) to demonstrate the effectiveness of our method. For MMU, besides unified models, we also compare UniCMs with VLMs (e.g., Emu3-Chat (Wang et al., 2024b), Qwen-VL (Bai et al., 2023)) in terms of both inference speed and accuracy, where the speed is measured on an RTX 4090 GPU.

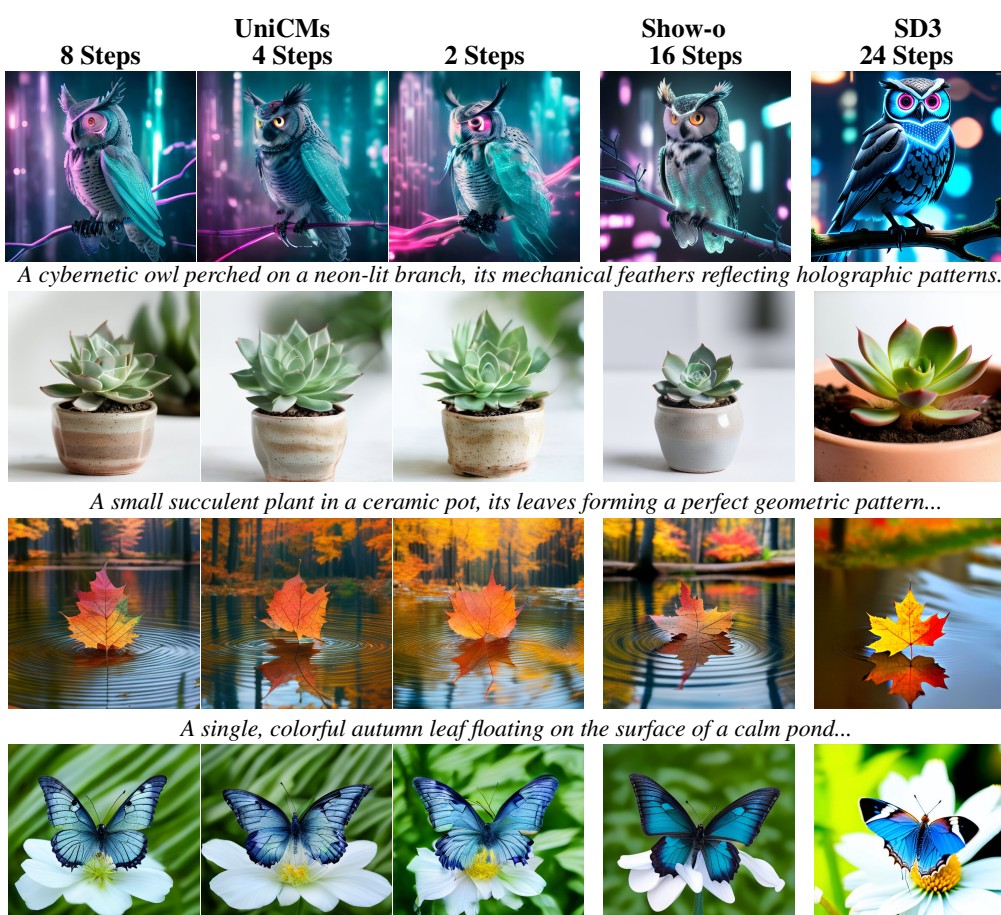

| UniCMs | | | Show-o | SD3 |
| 8 Steps | 4 Steps | 2 Steps | 16 Steps | 24 Steps |

*A cybernetic owl perched on a neon-lit branch, its mechanical feathers reflecting holographic patterns...*

*A small succulent plant in a ceramic pot, its leaves forming a perfect geometric pattern...*

*A single, colorful autumn leaf floating on the surface of a calm pond...*

*A blue butterfly resting on a white flower petal, its wings fully open to display vibrant patterns...*

Figure 4: **Comparison between UniCMs, Show-o, and SD3 in T2I generation at the resolution of 512 × 512.** Show-o is shown at 16 steps (using CFG), while UniCMs demonstrates performance at 8, 4, and 2 steps. SD3 results are included for comparison with UniCMs.

**Quantitative Results.** Table 1 shows the detailed results for T2I generation task. We observe that in 2-8 step sampling, UniCMs significantly outperform Emu3-Gen (Wang et al., 2024b), SDXL (Podell et al., 2023), SDXL-Turbo (Sauer et al., 2024), Hyper-SD3 (Ren et al., 2024) and Chameleon (Team, 2024a) on GenEval benchmark, without using CFG. Remarkably, UniCMs use approximately $1/8$ of the inference time of SD3 (Esser et al., 2024) and outperform both Hyper-SD3 (Ren et al., 2024) and SDXL-Turbo (Sauer et al., 2024) within similar inference time, highlighting the superior computational efficiency of UniCMs models. In the Appendix C, Table 7 reports the comprehensive performance of UniCMs and Show-o (Xie et al., 2024a) with an equal number of sampling steps, displaying the advantage of UniCMs in low-step generation scenarios. Besides, we can observe that UniCMs clearly outperform UniCMs* in Appendix C, Table 7, demonstrating the efficacy of the second training stage. Additionally, we demonstrate that CFG can further enhance UniCMs performance for image generation task with 4-16 step sampling in the Appendix C, Table 6.

Table 2 shows the performance of UniCMs in MMU tasks. We evaluate the text token generation speed on NoCaps (Agrawal et al., 2019), showing that UniCMs is on average 1.5× faster than Show-o (Xie et al., 2024a) while maintaining competitive performance. Besides, we notice that UniCMs outperform Show-o on MMMU (Yue et al., 2024) and ScienceQA-IMG (Lu et al., 2022). The slight drop on NoCaps and Flickr30K captioning reflects a speed–performance trade-off. We attribute this to two factors: (1) multi-step sampling in consistency models accumulates errors from overlapping time intervals, as noted in prior work (Kim et al., 2023); (2) UniCMs is trained on far fewer data

| Set. | #IT ↓ | POPE ↑ | MME ↑ | IR ↑ | CS ↑ |
|---|---|---|---|---|---|
| 4 | **10.57** | 72.6 | **803.4** | **0.586** | **0.307** |
| 2 | 12.48 | 69.8 | 595.8 | 0.500 | 0.306 |
| 1 | 11.71 | **74.1** | 675.3 | 0.270 | 0.304 |

Table 4: **Ablation on segment number.** #IT means the number of iterations required by parallel decoding to decode 16 text tokens.

| Set.$(\beta, \gamma)$ | #IT ↓ | POPE ↑ | MME ↑ | IR ↑ | CS ↑ |
|---|---|---|---|---|---|
| (0, 0) | **2.85** | 0.0 | 4.91 | -2.278 | 0.184 |
| (10, 50) | 12.71 | **74.8** | 798.4 | 0.483 | **0.307** |
| (20, 100) | 10.57 | 72.6 | **803.4** | **0.586** | **0.307** |

Table 5: **Ablation on the regularization coefficients in the total loss.**

(e.g., 120k COCO prompts) than its teacher, which uses millions of image–text pairs. Distillation with richer MMU trajectories may alleviate this gap.

**Qualitative Results.** Figure 4 compares image generation models across different sampling steps. UniCMs can produce clear, high-quality images in only 2–4 steps without CFG, achieving visual quality comparable to Show-o (Xie et al., 2024a) and SD3 (Esser et al., 2024), which require dozens of steps. Additional UniCM results in Figure 1 further demonstrate effective sampling with few steps. Figure 3 illustrates UniCM text sampling trajectories for several MMU cases. UniCMs predict 16 tokens within fewer than 10 iterations by generating multiple successive tokens per iteration and correctly anticipating later tokens.

We also present UniCM performance in image inpainting and extrapolation in Appendix B (Figures 5 and 6), where both tasks are completed in four steps without extra training.

### 4.3 ABLATION STUDIES

To analyze the influence of each part, we conduct a comprehensive ablation study with an image resolution of 256. Unless otherwise specified, we report the results after the first training stage (i.e., UniCMs*), and the T2I generation is done with 4 sampling steps.

**Number of Segments.** We study the influence of segments on UniCMs. As shown in Table 4, models trained in two segments and without trajectory segmentation (i.e., using one segment) can exhibit a suboptimal performance and a degraded

| Steps | Top-k | HPS ↑ | IR ↑ | CS ↑ |
|---|---|---|---|---|
| 4 | - | 0.245 | 0.621 | 0.306 |
| 4 | 200 | **0.252** | **0.706** | **0.309** |
| 2 | - | 0.216 | 0.027 | 0.291 |
| 2 | 10 | **0.240** | **0.529** | **0.306** |

Table 3: **Comparison on sampling strategy at the image resolution of 256.** "-" denotes standard multinomial sampling.

acceleration effect. This result reflects the effectiveness of our trajectory segmentation strategy for improving convergence speed and model performance.

**Regularization.** As shown in Table 5, training without regularization constraints (i.e., $\beta = 0, \gamma = 0$) tends to make the model collapse rapidly. Besides, smaller regularization weights can lead to inferior performance, highlighting the importance of regularization in constraining the distribution of UniCMs in training.

**Top-k Sampling.** Table 3 shows the results with different sampling strategies for T2I. We observe that top-k significantly improves the performance of UniCMs on 2-step and 4-step sampling. This is probably because there is high uncertainty in the output distribution of UniCMs.

## 5 CONCLUSIONS AND LIMITATIONS

In this paper, we introduce UniCMs, a unified consistency model family for multimodal generation and understanding. UniCMs adopt a unified denoising perspective for both text and image generation. They are trained via an adapted consistency distillation approach on collected multimodal trajectories, learning to map any point on the trajectory to the same endpoint. The unified training objective empowers UniCMs to deliver strong performance with significantly fewer steps across both multimodal generation and understanding tasks. For future work, we plan to scale our model on more advanced multimodal trajectories to further improve the performance of UniCMs.

## 6 ETHICS STATEMENT

This work does not involve human subjects, animal experiments, or sensitive data. Therefore, it does not raise any ethical concerns.

## 7 REPRODUCIBILITY STATEMENT

To promote reproducibility, we plan to release the source code for both training and inference of the proposed method in the future. The released code and accompanying instructions will enable other researchers to reproduce our results and build upon this work.

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

## A ABLATION STUDIES RESULTS

### A.1 THE USE OF LARGE LANGUAGE MODELS

In this work, large language models (LLMs) were employed solely as writing assistants. Their role was limited to polishing the language, improving clarity, and refining the overall readability of the manuscript. No part of the conceptual development, experimental design, data analysis, or interpretation of results relied on LLMs.

## B INPAINTING AND EXTRAPOLATION

Figure 5 shows that UniCMs can efficiently fill in missing parts of an image with high quality in just 2 to 4 steps, based on the given prompt. Meanwhile, Figure 6 demonstrates that UniCMs can smoothly complete image extrapolation in just 4 steps.

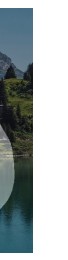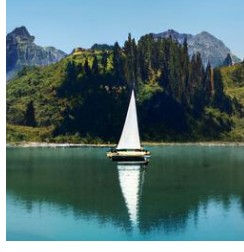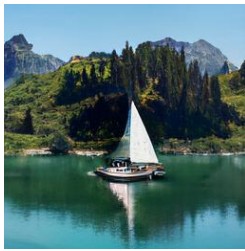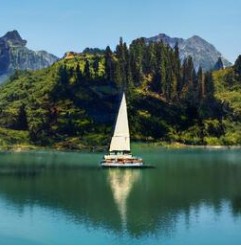

prompt: In the distance, a small white sailboat was parked between the mountains and the water.

Figure 5: **Visualization of image inpainting by UniCMs on 256 resolution.** From left to right are the 2, 4, and 8 steps sampling.

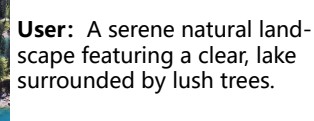

**User:** The mountains and jungles are covered with thin mist.

**User:** A serene natural landscape featuring a clear, lake surrounded by lush trees.

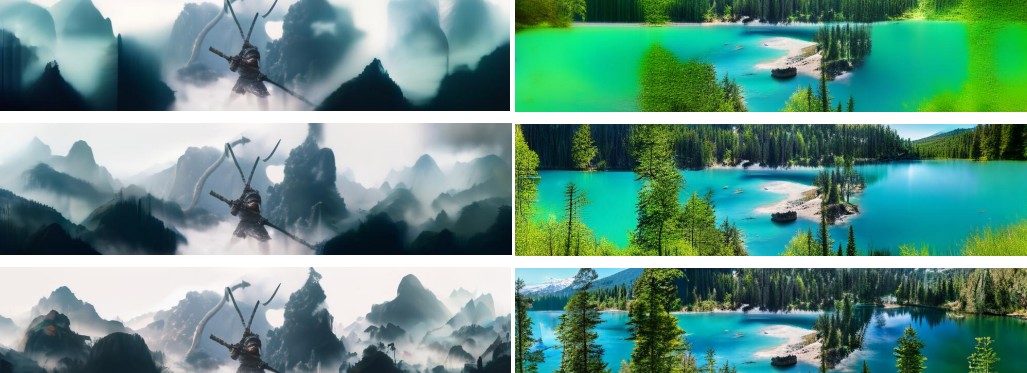

Figure 6: **Visualization of image extrapolation by UniCMs on 256 resolution.** From top to bottom are the 2, 4, and 8 steps sampling.

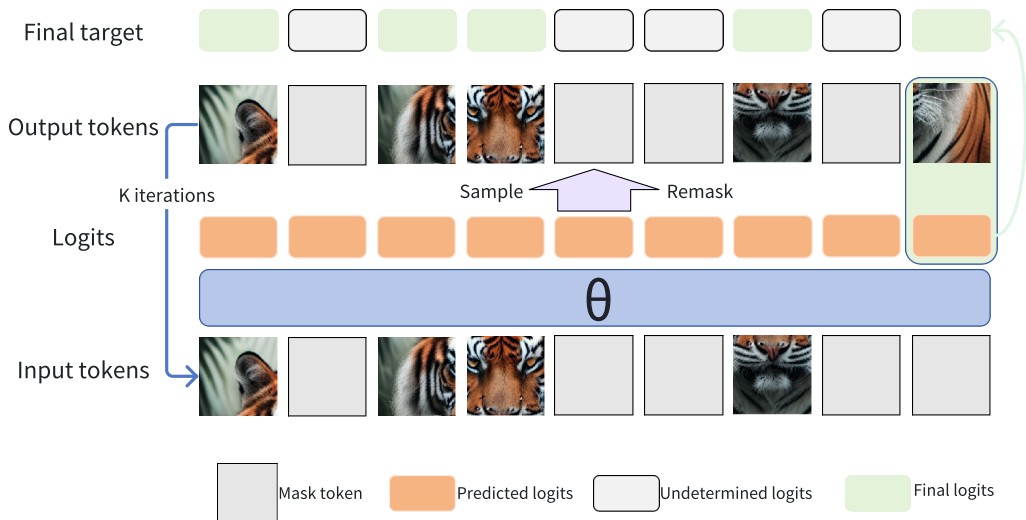

Figure 7: **Visualization of regularization label for image trajectory distillation.** For each iteration, we only record the logits of the region converted from the mask to the image token, and finally concatenate them into the regularization logits label. We abuse the $\theta$ to denote the mask diffusion models here.

| Model | Steps | CFG | HPS ↑ | IR ↑ | CS ↑ |
|---|---|---|---|---|---|
| UniCMs | 16 | 0 | **0.258** | 0.752 | **0.310** |
| | | 1 | **0.258** | **0.816** | **0.310** |
| | 8 | 0 | **0.255** | 0.738 | **0.309** |
| | | 1 | **0.255** | **0.782** | **0.310** |
| | 4 | 0 | **0.252** | 0.706 | **0.309** |
| | | 1 | **0.252** | **0.731** | **0.309** |
| | 2 | 0 | **0.240** | **0.529** | **0.306** |
| | | 1 | 0.235 | 0.420 | 0.302 |
| Show-o | 16 | 0 | 0.174 | -1.097 | 0.272 |
| | | 10 | **0.254** | **0.739** | **0.310** |
| | 8 | 0 | 0.181 | -0.916 | 0.276 |
| | | 10 | **0.249** | **0.665** | **0.308** |
| | 4 | 0 | 0.178 | -0.877 | 0.276 |
| | | 10 | **0.228** | **0.219** | **0.301** |
| | 2 | 0 | 0.159 | -1.661 | 0.234 |
| | | 10 | **0.169** | **-1.257** | **0.254** |

Table 6: **Results with different CFG on 256 resolution.** A proper CFG can enhance the performance of Show-o and UniCMs.

## C  SETTINGS OF CFG

As shown in Table 6, the appropriate use of CFG further enhances the sampling performance of UniCMs, particularly for sampling steps of 4 or more. Additionally, the performance of Show-o drops significantly without CFG, resulting in images that lack semantic information.

## D  REGULARIZATION LOSS DETAILS

The regularization loss for text trajectories is straightforward to compute because we only need $p_\phi$ to fit the endpoint text tokens $\mathbf{v}^K$. However, directly employing sampled images for the regularization loss of image trajectories degrades quality. This degradation arises because sampling images along

| Steps | Model | CFG | GenEval ↑ | | | | | | | HPS ↑ | IR ↑ | CS ↑ |
|---|---|---|---|---|---|---|---|---|---|---|---|---|
| | | | AVG | TO | CT | P | CL | SO | CA | | | |
| 16 | Show-o | 10 | **0.674** | 0.823 | 0.647 | 0.288 | 0.838 | 0.984 | 0.463 | **0.277** | **0.992** | **0.318** |
| | Show-o | 5 | 0.672 | 0.778 | 0.666 | 0.293 | 0.835 | 0.991 | 0.468 | 0.270 | 0.885 | **0.318** |
| | UniCMs* | 0 | 0.649 | 0.793 | 0.644 | 0.253 | 0.809 | 0.956 | 0.440 | 0.266 | 0.768 | 0.315 |
| | UniCMs | 0 | 0.646 | 0.818 | 0.597 | 0.218 | 0.827 | 0.984 | 0.430 | 0.273 | 0.925 | **0.318** |
| 8 | Show-o | 10 | 0.578 | 0.631 | 0.519 | 0.235 | 0.811 | 0.991 | 0.280 | 0.257 | 0.672 | 0.313 |
| | Show-o | 5 | 0.580 | 0.647 | 0.584 | 0.225 | 0.766 | 0.984 | 0.275 | 0.255 | 0.632 | 0.313 |
| | UniCMs* | 0 | **0.642** | 0.788 | 0.631 | 0.253 | 0.787 | 0.981 | 0.413 | 0.264 | 0.800 | 0.315 |
| | UniCMs | 0 | 0.638 | 0.813 | 0.541 | 0.250 | 0.814 | 0.991 | 0.420 | **0.273** | **0.963** | **0.318** |
| 4 | Show-o | 10 | 0.353 | 0.237 | 0.325 | 0.095 | 0.540 | 0.863 | 0.060 | 0.197 | -0.560 | 0.283 |
| | Show-o | 5 | 0.396 | 0.298 | 0.334 | 0.158 | 0.572 | 0.925 | 0.088 | 0.207 | -0.300 | 0.294 |
| | UniCMs* | 0 | 0.596 | 0.692 | 0.553 | 0.218 | 0.758 | 0.978 | 0.375 | 0.249 | 0.633 | 0.312 |
| | UniCMs | 0 | **0.625** | 0.770 | 0.553 | 0.245 | 0.806 | 0.978 | 0.398 | **0.269** | **0.934** | **0.318** |
| 2 | Show-o | 10 | 0.181 | 0.025 | 0.131 | 0.008 | 0.327 | 0.588 | 0.008 | 0.140 | -1.756 | 0.246 |
| | Show-o | 5 | 0.251 | 0.051 | 0.188 | 0.038 | 0.442 | 0.778 | 0.010 | 0.152 | -1.456 | 0.260 |
| | UniCMs* | 0 | 0.459 | 0.407 | 0.422 | 0.148 | 0.668 | 0.925 | 0.185 | 0.201 | -0.259 | 0.295 |
| | UniCMs | 0 | **0.557** | 0.614 | 0.478 | 0.180 | 0.793 | 0.972 | 0.305 | **0.247** | **0.680** | **0.312** |

Table 7: **Comparison of T2I performance at the resolution of 512 × 512 based on GenEval, HPS, IR, and CS.** AVG: average, TO: Two Object, CT: Counting, P: Position, CL: colors, SO: Single Object, CA: Color Attr.

| Steps | Model | CFG | GenEval ↑ | | | | | | | HPS ↑ | IR ↑ | CS ↑ |
|---|---|---|---|---|---|---|---|---|---|---|---|---|
| | | | AVG | TO | CT | P | CL | SO | CA | | | |
| 16 | Show-o | 10 | **0.591** | 0.692 | 0.478 | 0.165 | 0.859 | 0.978 | 0.378 | 0.254 | 0.739 | **0.310** |
| | Show-o | 5 | 0.571 | 0.631 | 0.469 | 0.155 | 0.846 | 0.994 | 0.333 | 0.253 | 0.642 | 0.309 |
| | UniCMs* | 0 | 0.543 | 0.593 | 0.447 | 0.130 | 0.814 | 0.953 | 0.323 | 0.251 | 0.586 | 0.307 |
| | UniCMs | 0 | 0.562 | 0.689 | 0.366 | 0.140 | 0.814 | 0.991 | 0.373 | **0.258** | **0.752** | **0.310** |
| 8 | Show-o | 10 | 0.540 | 0.578 | 0.428 | 0.145 | 0.838 | 0.969 | 0.285 | 0.249 | 0.665 | 0.308 |
| | Show-o | 5 | 0.530 | 0.558 | 0.441 | 0.133 | 0.825 | 0.972 | 0.255 | 0.247 | 0.602 | 0.308 |
| | UniCMs* | 0 | 0.518 | 0.518 | 0.400 | 0.123 | 0.809 | 0.972 | 0.285 | 0.250 | 0.597 | 0.307 |
| | UniCMs | 0 | **0.552** | 0.669 | 0.353 | 0.128 | 0.817 | 0.963 | 0.385 | **0.255** | **0.738** | **0.309** |
| 4 | Show-o | 10 | 0.425 | 0.333 | 0.334 | 0.100 | 0.700 | 0.950 | 0.135 | 0.228 | 0.219 | 0.301 |
| | Show-o | 5 | 0.429 | 0.351 | 0.369 | 0.078 | 0.707 | 0.947 | 0.120 | 0.228 | 0.225 | 0.302 |
| | UniCMs* | 0 | 0.504 | 0.513 | 0.375 | 0.130 | 0.787 | 0.962 | 0.257 | 0.245 | 0.586 | 0.307 |
| | UniCMs | 0 | **0.523** | 0.664 | 0.303 | 0.103 | 0.801 | 0.959 | 0.308 | **0.252** | **0.706** | **0.309** |
| 2 | Show-o | 10 | 0.206 | 0.046 | 0.140 | 0.033 | 0.330 | 0.678 | 0.010 | 0.169 | -1.257 | 0.254 |
| | Show-o | 5 | 0.229 | 0.068 | 0.122 | 0.023 | 0.378 | 0.763 | 0.020 | 0.182 | -0.917 | 0.263 |
| | UniCMs* | 0 | 0.439 | 0.358 | 0.313 | 0.075 | 0.755 | 0.941 | 0.193 | 0.224 | 0.174 | 0.302 |
| | UniCMs | 0 | **0.494** | 0.530 | 0.334 | 0.093 | 0.787 | 0.959 | 0.260 | **0.240** | **0.529** | **0.306** |

Table 8: **Comparison of 256 × 256 T2I performance on GenEval, HPS, IR, and CS.** UniCMs* refers to the model after the first stage of training. AVG: average, TO: Two Object, CT: Counting, P: Position, CL: colors, SO: Single Object, CA: Color Attr.

a fixed trajectory under a greedy strategy diminishes both their diversity and quality. Moreover, the T2I model's distribution encapsulates rich information, which is inherently diminished during the sampling process due to information loss. To address this, we propose constructing regularized logits labels by capturing the T2I model's distribution at each sampling step. As illustrated in Figure 7, we initialize a global logits target as an all-zero tensor. During the iteration of the trajectory $\mathbf{u}^k$, we focus on regions transitioning from mask to image tokens, populating the final target with the

| Method | Decoding | tokens/s ↑ | POPE ↑ | MMMU ↑ | Flickr30K ↑ | NoCaps ↑ |
|---|---|---|---|---|---|---|
| Show-o | AR | 40.3 | **83.2** | 24.6 | **24.9** | **29.4** |
| | Jacobi | 36.9 | **83.2** | 24.6 | **24.9** | **29.4** |
| UniCMs* | Jacobi | 49.9 | 81.8 | 25.4 | 23.5 | 28.1 |
| UniCMs | Jacobi | **61.1** | 78.4 | **26.3** | 22.2 | 26.5 |

Table 9: **Comparison of I2T performance at the resolution of 512 × 512 on multiple benchmarks.** Note that Flickr30K and NoCaps evaluate the ability of image description, and POPE and MMMU measure question-answering ability.

corresponding predicted logits for these regions. Through this iterative procedure, we synthesize a complete logits target, enabling the computation of $\mathcal{L}_{REG}^{u}$. If a segmentation strategy is adopted, the missing portions of the logits target can be populated with the final predicted logits at the segmentation endpoints. This produces a complete regularization label.

# E  SEGMENTATION DETAILS

Direct learning of consistency across an entire trajectory is challenging for models and often leads to convergence difficulties. Therefore, we propose applying a segmentation strategy to the multimodal denoising trajectory. Specifically, we evenly divide the trajectory into several segments and enforce consistency constraints between a randomly selected point within a segment and the endpoint of that segment, rather than the endpoint of the entire trajectory. For image trajectories, the regularization logits labels constructed from segment endpoints are incomplete. We address this by filling the missing parts with the logits predicted from the last iteration of that segment. We only compute the consistency loss in the masked regions of the segment endpoints and the regularization loss in the masked regions of the randomly selected points. For text trajectories, we continue to use noise-free text as the regularization constraint, introducing segmentation only in the consistency loss. Through ablation studies in Section 4.3, we demonstrate that this objective is more amenable to learning, facilitating model convergence toward the target and enhancing the effectiveness of acceleration.

# F  TRAINING DETAILS AND RESULTS OF 256 RESOLUTION

For 256 resolution, we separate the training process into two stages. In the first stage, we get image trajectories with a CFG scale of 10 and $K = 16$. We split each trajectory into 4 segments to train the consistency model, denoted as UniCMs*. In the second stage, we collect image trajectories using UniCMs*. We sample image trajectories with a CFG scale of 1.5, $K = 8$, and the number of segments as 2. The text trajectories are collected similarly. We employ Jacobi decoding to iteratively produce 16 tokens in each round to finally form lengthy text, which proves to yield good acceleration performance while preserving the generative modeling capabilities (Kou et al., 2024a). In terms of loss coefficients, we set $\alpha = 10$ according to the relative values of the losses, set $\beta = 20$ and $\gamma = 100$ according to the ablation study in Table 5, and set $\delta = 2$ following (Xie et al., 2024a). We use an AdamW optimizer and 8 RTX 4090 GPUs to train each stage for 18 hours, with a constant learning rate of $10^{-5}$.

| Method | Decoding | tokens/s ↑ | POPE ↑ | MME ↑ | MMMU ↑ | Flickr30K ↑ | NoCaps ↑ |
|---|---|---|---|---|---|---|---|
| Show-o | AR | 41.8 | **73.8** | **948.4** | 25.1 | **20.8** | **25.8** |
| | Jacobi | 38.2 | **73.8** | **948.4** | 25.1 | **20.8** | **25.8** |
| UniCMs* | Jacobi | 61.3 | 72.6 | 803.4 | **27.0** | 19.8 | 23.8 |
| UniCMs | Jacobi | **64.5** | 73.2 | 872.4 | 25.8 | 19.2 | 23.0 |

Table 10: **Comparison of 256 × 256 MMU performance on multiple benchmarks.** Note that Flickr30K and NoCaps evaluate the ability of image description, and POPE, MME, and MMMU measure question-answering ability.

Table 8 and Table 10 show the performance of UniCMs on T2I and MMU tasks at 256-resolution respectively. It can be observed that UniCMs can also achieve the effect of 8 steps of the original model in 4-step sampling without CFG in 256-resolution image generation, and also achieves about 1.5 times acceleration in 256-resolution image understanding.

# G   ABLATION STUDY ON BLOCK SIZE FOR IMAGE-TO-TEXT GENERATION

To investigate the impact of the parallel decoding block size on the inference speed of UniCMs, we conduct an ablation study on the NoCaps dataset. As parallel decoding introduces overhead, its efficiency is highly dependent on the length of the generated sequence. We therefore evaluate two distinct scenarios: a short caption generation task (e.g., 10-30 tokens) and a long caption generation task (e.g., 80-120 tokens). We measure the throughput in Tokens Per Second (TPS) and the average number of iterations required to decode a single block.

The results, presented in Table 11 and Table 12, reveal a clear trade-off.

| Block Size | TPS ↑ | Avg. Iterations/Block ↓ |
|---|---|---|
| 16 | 61.1 | 10.6 |
| 32 | 43.1 | 19.4 |
| 64 | 23.5 | 35.6 |

Table 11: Performance on **short** caption generation tasks with varying block sizes. Larger block sizes increase overhead and reduce overall throughput for short sequences.

| Block Size | TPS ↑ | Avg. Iterations/Block ↓ |
|---|---|---|
| 16 | 80.3 | 10.8 |
| 32 | 90.1 | 18.5 |
| 64 | 92.6 | 35.6 |

Table 12: Performance on **long** caption generation tasks with varying block sizes. For longer sequences, larger block sizes significantly improve throughput.

As indicated, for short generation tasks like standard image captioning, increasing the block size from 16 to 64 significantly raises the generation overhead (from 10.6 to 35.6 iterations), thereby reducing the effective throughput (TPS). Conversely, for long generation tasks, the overhead of parallel decoding is amortized over a longer sequence, and larger blocks substantially improve throughput. With a block size of 64, the throughput reaches 92.6 TPS, achieving a significant **2.3× speedup** compared to the 40.3 TPS of the autoregressive baseline (Show-o).

The 1.5× speedup reported in the main manuscript corresponds to the more common short captioning scenario (block size 16), which represents a conservative estimate of the acceleration capability of UniCMs.

# H   ADDITIONAL IMAGE RESULTS

Figure 8 and Figure 9 show the image generation results for 512 and 256 resolutions respectively. UniCMs can generate high-quality images with rich details using only 2 to 4 sampling steps and without CFG.

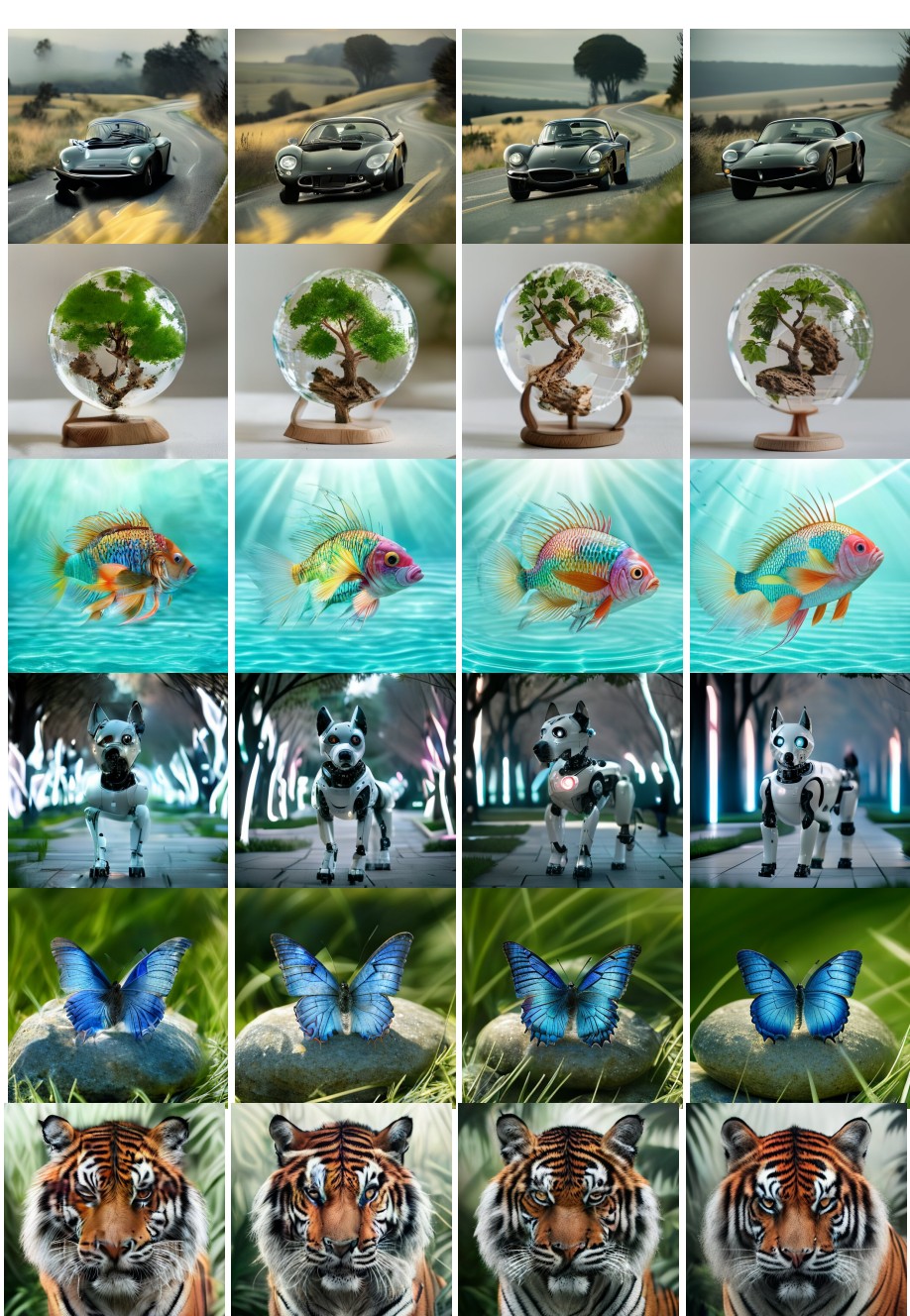

Figure 8: **512 × 512 images generated by UniCMs.** From left to right, the images are generated by UniCMs in 2, 4, 8 and 16 sampling steps without CFG.

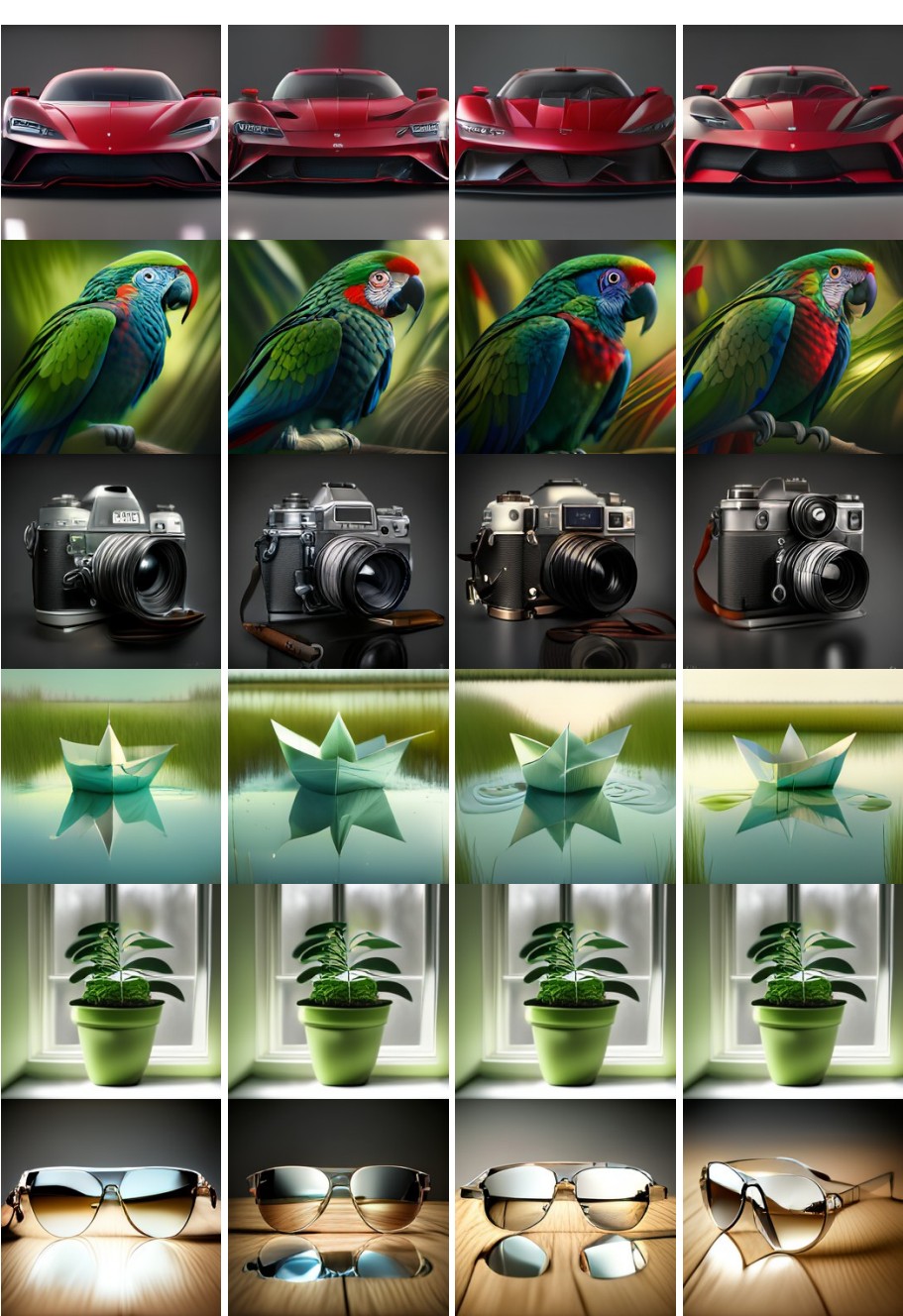

Figure 9: **256 × 256 images generated by UniCMs.** From left to right, the images are generated by UniCMs in 2, 4, 8 and 16 sampling steps without CFG.