# OpenReview forum: "UniCMs: A Unified Consistency Model For Efficient Multimodal Generation and Understanding"
_ICLR.cc/2026/Conference — ICLR 2026 Conference Withdrawn Submission_

### Official Review · Reviewer_ZYbn · 2025-10-31

**Soundness:** 3
**Presentation:** 4
**Contribution:** 2
**Rating:** 4
**Confidence:** 2

**Summary:**

This paper presents UniCMs, a consistency model framework designed to accelerate multimodal generation and understanding, unifying text-to-image and image-to-text tasks. The approach leverages discrete diffusion for images and parallel decoding for text, with consistency mapping and trajectory segmentation to improve efficiency. Experiments show that UniCMs achieve faster inference compared to existing baselines, with competitive results on standard benchmarks.

**Strengths:**

1. Practical speedup: UniCMs deliver significant acceleration, which is valuable for real-world deployment.
2. Comprehensive evaluation: The experiments are thorough, with strong baselines and ablations.
3. Clarity: The paper is clearly written and well-illustrated.

**Weaknesses:**

1. Limited Technical Contribution: The main advance is in speed, not in new algorithms or modeling paradigms. The technical novelty is incremental.
2. Performance not state-of-the-art: UniCMs do not outperform the best existing models on all benchmarks; there is a clear trade-off between speed and accuracy.
3. Initialization from Show-o: The framework is initialized with Show-o’s architecture and parameters, and the resulting performance is only comparable to Show-o, which further limits the originality and significance of the contribution.
4. Scope of evaluation: The evaluation is focused on standard benchmarks; broader applicability is not explored.
5. Teacher model dependence: The reliance on teacher models for trajectory collection may limit generalization.

**Questions:**

1. Novelty clarification: Please clarify what is genuinely new in your approach beyond efficiency improvements and adaptation of consistency models.
2. Performance limitation: Are there strategies to further close the gap on tasks where UniCMs underperform without sacrificing speed? Can you provide more analysis on the speed–accuracy trade-off?
3. Initialization impact: Given that UniCMs are initialized from Show-o and achieve similar performance, what unique advantages does your approach offer beyond speed?
4. Broader applicability: Can UniCMs be extended to other modalities or tasks? What are the challenges?
5. Teacher model sensitivity: How sensitive are UniCMs to the choice of teacher model for trajectory collection?

---

### Official Review · Reviewer_5mUr · 2025-11-01

**Soundness:** 1
**Presentation:** 3
**Contribution:** 2
**Rating:** 2
**Confidence:** 3

**Summary:**

The paper proposes a way to distill models like Show-O that leverage an autoregressive component for text and masked diffusion for the image component. Specifically the present a generalization of consistency models for masked diffusion and leverage this to have a unified model that is capable of generating images in less steps while preserving text generation as well as understanding.

**Strengths:**

- The paper is well written and easy to follow
- The numerical results are very good, their distilled version of Show-o is almost as capable as the original model but cutting down the computational cost significantly.
- The experimental study is comprehensive with good ablations studies

**Weaknesses:**

- The consistency loss for discrete diffusion is not very consistent. The distilled probability distributions would have to include the correlations between different tokens, however, in the current presentation, these are dropped, and it is unclear how the model is managing to actually distill things if tokens are being generated independently. A work that has studied this problem is [1] from which it can be seen the importance of learning such correlations. Without further explanation of this point the current paper is hard to assess.
- If the above can be explained, how is the current distillation method different from [1]
-



[1] Hayakawa, Satoshi, et al. "Distillation of discrete diffusion through dimensional correlations." arXiv preprint arXiv:2410.08709 (2024).

**Questions:**

- Why doing consistency models on works that leverage continuous diffusion as opposed to discrete diffusion as in this setting the consistency loss would apply natively? Some examples of these work include [1],[2],[3]



[1] Li, Shufan, et al. "Omniflow: Any-to-any generation with multi-modal rectified flows." Proceedings of the Computer Vision and Pattern Recognition Conference. 2025.

[2] Rojas, Kevin, et al. "Diffuse Everything: Multimodal Diffusion Models on Arbitrary State Spaces." arXiv preprint arXiv:2506.07903 (2025).

[3] Ma, Yiyang, et al. "Janusflow: Harmonizing autoregression and rectified flow for unified multimodal understanding and generation." Proceedings of the Computer Vision and Pattern Recognition Conference. 2025.

---

### Official Review · Reviewer_u6XA · 2025-11-01

**Soundness:** 3
**Presentation:** 3
**Contribution:** 2
**Rating:** 4
**Confidence:** 4

**Summary:**

This paper presents accelerated unified multimodal models from the perspective of the consistency model. By applying consistency distillation to the parallel decoding of the language model and discrete diffusion, the distilled unified models achieve comparable performance to the state-of-the-art approaches and significantly accelerate the inference speed.

**Strengths:**

1. This paper is well-organized and easy to read and follow.
2. The overall motivation is clear and logical.
3. The distilled model achieves significant acceleration while preserving the multimodal understanding and generation performance.

**Weaknesses:**

1. I thought it was necessary to study the acceleration in unified multimodal models. I was wondering if there are unique challenges in this area instead of directly adopting the distillation approaches well-studied in large language models and consistency models.

2. It looks like there are significant performance drops in several metrics in the T2I and MMU evaluations.

3. Is it possible to wrap the proposed pipeline into a general acceleration, as a lot of unified multimodal models have been successively proposed? It would significantly enhance the impact and insight of this work.

**Questions:**

See weaknesses. I would like to adjust my initial rating according to the authors' response.

---

### Note · Authors · 2026-01-05

**Comment:**

Dear Program Chairs and Reviewers, we are writing to formally withdraw our submission #24596, titled "UniCMs: A Unified Consistency Model For Efficient Multimodal Generation and Understanding," as we have decided to further refine the methodology and conduct additional experiments based on your valuable feedback. We sincerely appreciate the professional insights and the significant time you have invested in reviewing our work, and we look forward to submitting a strengthened version in the future.

**Withdrawal Confirmation:**

I have read and agree with the venue's withdrawal policy on behalf of myself and my co-authors.